# Effect of the Temperature–Humidity Index on the Productivity of Dairy Cows and the Correlation between the Temperature–Humidity Index and Rumen Temperature Using a Rumen Sensor

**DOI:** 10.3390/ani14192848

**Published:** 2024-10-03

**Authors:** Ki Taeg Nam, Nackhoon Choi, Youngjun Na, Yongjun Choi

**Affiliations:** 1Department of Animal Science and Biotechnology, Hankyong National University, Anseong 17579, Republic of Korea; ktnam@hknu.ac.kr; 2Department of Animal Science and Technology, Konkuk University, Seoul 05029, Republic of Korea; loveaji@hanmail.net (N.C.); ruminoreticulum@gmail.com (Y.N.); 3Animal Data Laboratory, Antller Inc., Seoul 05029, Republic of Korea

**Keywords:** temperature–humidity index, milk production, milk composition, rumen sensor, dairy cows

## Abstract

**Simple Summary:**

This study evaluated the effects of heat stress on the productivity of dairy cows. During the experimental period, THI values ranged from 46.9 to 81.0, with the most severe heat stress occurring in August. A decrease in productivity was observed when the THI exceeded 72. It was found that milk production took about one month to recover after the high-temperature period ended. The THI ratio, calculated based on the daily THI distribution, showed a strong negative correlation with rumen temperature (− 0.744; *p* < 0.001). During periods of significant productivity decreases due to high-temperature stress, rumen temperatures were higher than 39.15 °C. Based on the rumen temperature and daily THI distribution, it is suggested that these findings could serve as foundational data for developing technologies to manage high-temperature environments for dairy cows.

**Abstract:**

This study was conducted to evaluate the effects of high-temperature stress on dairy cow productivity and the correlation between rumen sensors. The data were collected on the temperature, humidity, milk productivity, milk components, blood components, and rumen sensor data from 125 dairy cows during the experimental period (1 May 2020 to 30 October 2020). High-temperature stress of dairy cows was evaluated based on the temperature–humidity index (THI). The correlations between the high-temperature stress, productivity, and sensor data were analyzed using SAS and R programs. The THI ranged from 46.9 to 81.0 during the experimental period, and a significant decrease was observed in the milk production of dairy cows during August (*p* < 0.05). Milk production was evaluated to decrease by 1.8% because of high-temperature stress during the experimental period. There was a significantly high negative correlation between the THI ratio of day and rumen temperature (r = 0.744; *p* < 0.001). The other rumen sensor data did not show a significant correlation with the productivity of dairy cows. The results can be utilized as a guideline for managing temperature and humidity to maintain dairy cow productivity on farms in high-temperature stress conditions.

## 1. Introduction

Holstein generally refers to the North American breed, which has been selectively improved for the productivity of dairy products. The term “Friesian” refers to cattle of traditional European descent, characterized by their smaller body size. These animals are still raised for both dairy production and meat. The Holstein Friesian breed, primarily raised in South Korea, is similar to the large breed found in the United States, which emphasizes productivity. Holstein dairy cattle can endure low temperatures during cold winters; however, they are known to be adversely susceptible to high-temperature conditions [1]. High-temperature stress has been identified as a major factor reducing livestock productivity worldwide, leading to extensive research on its impact on nutrition [2,3,4,5].

Recent global warming, attributed to the increase in greenhouse gases, has led to rapidly rising atmospheric temperatures. These issues ultimately have a negative impact on dairy cow productivity. High-temperature stress in Holstein dairy cows negatively affects factors such as feed intake, milk production, milk fat, and somatic cell count [6,7]. Evaluating cattle heat stress using the temperature–humidity index (THI) is categorized as follows: THI values of 70–72 (Stress Threshold) are considered normal for heart rate, ruminal temperature, and surface temperature; THI values of 73–78 (mild–moderate) see an increase in heart rate, ruminal temperature, surface temperature, standing time, and water intake; THI values of 79–88 (moderate–severe) witness further increases in heart rate, ruminal temperature, surface temperature, stress hormones, and water intake, while feed intake decreases [8]; and lastly, THI values of 89–96 (severe) result in increases in heart rate, ruminal temperature, surface temperature, and water intake, along with decreased feed intake and risk of heat-related mortality [8]. When the THI exceeds 80, it is reported that milk production in Holstein cows decreases by 19% [9]. Additionally, the feed efficiency of livestock decreases [6] in regions experiencing high-temperature stress. High-temperature stress also reduces dry matter intake and energy supply, leading to negative energy balance (NEB) and subsequent weight loss [10]. Hormonal changes occur in livestock because of high-temperature stress, resulting in decreased secretion of growth hormones, catecholamines, and cortisol, lowering metabolic rates and heat production [11]. Bicarbonate is a source of carbon dioxide (CO_2_), a by-product gas produced when the body burns food for energy. High-temperature stress raises the body temperature of Holstein dairy cows, causing their respiration rates to increase as a compensatory mechanism. This, in turn, decreases the blood CO_2_ ratio, which results in a condition known as blood acidosis. In animals, to maintain normal CO_2_ levels in the blood, the kidneys reduce bicarbonate (HCO_3−_) secretion. This reduction in bicarbonate leads to decreased dry matter intake and reduced chewing activity. This reduction in chewing activity impacts the supply of saliva, which is essential for maintaining the pH environment in the rumen. A decreased supply of saliva leads to rumen acidosis. These phenomena occur in a chain reaction, ultimately affecting the overall dairy cow productivity.

Previous studies have primarily focused on the impact of high-temperature stress on dairy cattle productivity and health. With the recent advancement of internet of things (IoT) technology, various remote monitoring tools based on sensors have been introduced in the livestock industry, aiding management. These remote monitoring devices come in various forms, including ear tags, ankle attachments, rumen-inserted devices, and necklaces. These devices mainly provide data on animal estrus detection, rumination, activity, feed intake, and behavior. In South Korea, farmers are increasingly using such attachment-based devices for estrus detection and cattle management, and these data have recently increased. The rumen insertion sensor measures rumen temperature, pH, and phase changes, among other factors. It provides detailed individual information, including signs of estrus, drinking frequency, and activity levels. However, research on the relationship between high-temperature stress and data collected through IoT technology is limited.

Therefore, this study aims to investigate the impact of high-temperature stress during the summer season on Holstein dairy cow’s productivity and correlation utilizing a rumen sensor. Also, this study aims to provide foundational data and establish criteria for productivity recovery periods.

## 2. Materials and Methods

All research protocols were performed according to the Konkuk University Animal Care and Use Committee guidelines (approval number: KU18094). 

### 2.1. Animal Care 

This experiment utilized 125 Holstein Friesian lactating dairy cows (average milk yield, 32.9 ± 11.9; average parity, 2.3 ± 1.5; average days in milk, 153.5 ± 103.8) from Dansung Farm located in Seosan City, Chungcheongnam-do, South Korea, from 1 May 2020 to 30 October 2020. The dairy cow feed was provided in the form of a total mixed ration (TMR) and administered twice daily (Table 1). Feed, water, and mineral blocks were made available ad libitum. Milking was conducted twice daily at 0600 h in the morning and 1700 h in the evening, using a tandem mechanical milking system (Matatron 21, GEA Westfalisurge, Bönen, Germany). At the beginning of the experiment (28 April 2020), the cow barn floors were cleaned of manure, and sawdust bedding of at least 5 cm in depth was spread. Throughout the experimental period, the bedding, mixed with cattle manure, was replaced with fresh sawdust bedding three times. To provide an objective assessment of temperature and humidity conditions, the fan speed and direction were maintained similarly throughout the experiment period. Fans were only operated from 1000 to 1800 h. The early lactation period was defined as 1 to 70 days (n = 75), mid-lactation was defined as 71 to 140 days (n = 80), and late lactation was defined as over 140 days (n = 286) until 305 days. 

### 2.2. Acquisition of Weather Data and Calculation of the Temperature–Humidity Index (THI)

Temperature and humidity values within the barn were measured using small-sized temperature and humidity data loggers (MHB-382SD, Lutron Electronic, PA, USA) attached to the pillars at a height similar to that of the cow’s bodies. These data loggers were set to record temperature and humidity automatically at 30 min intervals. External temperature and humidity data outside the farm were obtained from the Korea Meteorological Administration’s daily weather data for the Seosan area. To ensure data accuracy, measurements from the data loggers were compared with the data from the Korea Meteorological Administration.

The THI relevant to cattle was calculated using the prediction model for the THI provided by the NRC method [8]. The THI calculation formula is as follows:THI = (1.8 × T_db_ + 32) − [(0.55 − 0.0055 × RH) × (1.8 × T_db_ − 26.8)]
where T_db_ represents the dry bulb temperature (ºC) and RH represents the average relative humidity (%). Based on these criteria (NRC 2001), the THI was calculated from the measured temperature and humidity data, and its impact on the productivity, milk composition, and behavioral changes in the dairy cows was assessed. The THI ratio refers to the time proportion of stress levels categorized as below threshold, mild to moderate, moderate to severe, and severe over a 24 h period. 

### 2.3. Milk Composition Analysis

To evaluate milk production based on the THI, cows with irregular lactation cycles lasting less than 10 days and daily milk productions of less than 10 L were excluded from the analysis. For milk component analysis, milk samples were collected once a month from all milking cows using a tandem mechanical milking system (Matatron 21, GEA Westfalisurge, Bönen, Germany). Milk components, including milk fat, milk protein, milk solids-not-fat, milk urea nitrogen (MUN), and somatic cell count, were analyzed using an automated milk component analyzer (Milko-scan FT 6000, Foss electric Co., Hillerod, Denmark) at the Central Milk Component Analysis Laboratory of the Dairy Improvement Association.

### 2.4. Rumen Temperature, Estrus Index, and Activity

Rumen temperature and activity were monitored using a bolus-type rumen environmental monitoring sensor (Figure 1; Bolus pH plus, SMAXTEC animal care GmbH, Graz, Austria) inserted into the rumen of 75 randomly selected cows from the milking population. The sensor provided measurements for rumen temperature and activity at 10 min intervals. The Bolus sensor consists of a pH, temperature, and a gyro sensor. Rumen temperature was directly obtained from the temperature sensor. At the same time, activity was calculated based on phase changes in an animal’s movement derived from gyro sensor data using the company’s proprietary methods [12]. The pH data were unusable because of the breakdown of most sensors within one month in this study.

### 2.5. Feed Chemical Composition

The feed provided during the experimental period was analyzed for its general composition according to the methods outlined by AOAC (2005). This included dry matter (DM; AOAC official method 930.15), ash content (AOAC official method 930.15), crude protein (CP; AOAC official methods 930.15 and 990.03), ether extract (EE; AOAC official methods 930.15 and 2003.05), neutral detergent fiber (NDF; AOAC official method 942.05), and acid detergent fiber (ADF; AOAC official methods 930.15 and 973.18). Non-fiber carbohydrates (NFCs) were calculated as the difference between 100% and the sum of CP, EE, ash, and NDF.

### 2.6. Statistical Analysis

To investigate the impact of high-temperature stress on dairy cow productivity, the PROC MIXED procedure in the SAS package program (SAS, 2013; version 9.4, SAS Inc., Cary, NC, U.S.A.) was employed. The model used in the analysis by month is as follows:Y_ij_ = μ + T_i_ + P_j_ + E_ij(k)_
where μ represents the mean value, Ti represents the month, P_j_ represents parity, and E_ij_ represents the experimental error. The fixed variables included the monthly effect during experimental periods (May 01, 2020 to October 30, 2020), and the random variable considered parity. The significance of treatment groups was tested at the *p* < 0.05 level, and trends were assessed at the 0.05 ≤ *p* < 0.10 level using the CONTRAST and PDIFF options for repeated measurements and multiple comparisons. Milk productivity, milk composition, and sensor data were calculated as least squares mean for each animal by month and are presented in tables for comparison.

Data editing and visualization were conducted using the dplyr and tidyverse packages within the R package (R version 3.31, R Foundation, Tokyo, Japan). 

The milk production reduction evaluation was performed by removing the data for the period in which the THI exceeded 72 and comparing the difference between the linear trend line derived from the remaining data and the nonlinear trend line derived from the entire dataset. The regression model used in Figure 2 was
Y_ij_ = μ + T_i_ + E_i(j)_
where μ represents the mean value, Ti represents time effect, and E_ij_ represents the experimental error. Random effects were not considered. The linear regression was calculated using the ggplot2 package of the lm method, and the nonlinear regression was calculated using local regression smoothing splines using that of the gam method. Nonlinear trend lines were applied to the THI, rumen temperature, THI ratio, and heat index using the ggplot2 package of locally estimated scatterplot smoothing (losses) method [13].

## 3. Results

### 3.1. Temperature, Humidity, and THI

The variations in temperature, humidity, and the temperature–humidity index (THI) during the experiment conducted from May to October 2020 are presented in Table 1. The average temperature was the highest in August at 26.2 °C, followed by July, June, September, May, and October, with temperatures of 22.9, 22.1, 20.3, 16.8, and 13.3 ºC, respectively (*p* < 0.001). The lowest temperatures followed a similar pattern to the average temperatures, with August, July, June, September, May, and October registering temperatures of 23.8, 20.0, 18.3, 16.5, 12.3, and 7.8 °C, respectively (*p* < 0.001). The maximum temperature was recorded in August at 29.5 °C, followed by June and July at 27.4 and 26.9 °C, respectively, and then September, May, and October at 25.1, 22.3, and 19.6 °C, respectively (*p* < 0.001).

The average humidity was the highest in August at 86.3% (*p* < 0.05), followed by July, September, June, May, and October at 82.2, 78.4, 78.3, 73.6, and 68.7%, respectively (*p* < 0.05). The humidity in June and September did not significantly differ. The highest minimum humidity was observed in July and August at 71.8 and 63.7%, respectively, while May, June, and September exhibited similar humidity levels at 48.2, 52.7, and 54.6%, respectively, and October had the lowest humidity at 37.7% (*p* < 0.05). The maximum humidity showed varying significant patterns across months (Table 2, *p* = 0.002).

The average THI was the highest in August at 77.3, followed by July, June, September, May, and October at 71.6, 69.6, 67.1, 61.1, and 55.8, respectively (*p* < 0.05). The minimum THI was found in August at 73.6, with June, July, and September following closely at 67.6, 64.8, and 61.6, respectively, while May and October had the minimum THI at 54.3 and 46.9, respectively (*p* < 0.05). The maximum THI was found in August at 81.0, with June and July close behind at 64.8 and 67.6, respectively, while September, May, and October registering 73.0, 68.2, and 64.7, respectively (*p* < 0.05). 

### 3.2. Dry Matter Intake, Days in Milk, and Milk Production

Throughout the experimental period, the feed intake of dairy cows showed a significant decrease in August, with the lowest intake recorded at 36.4 kg/head/day (*p* < 0.001) (Table 3). The number of dry days varied by month, with June having the highest days in milk at 38.4, while August had the lowest at 31.7 (*p* < 0.05). In the mid-lactation period, August had the highest average days in milk at 109.7 (*p* < 0.05), while October had the lowest at 95.1 (*p* < 0.05), with the other months showing similar values. In the late lactation period, October had the highest average days in milk at 275.6, while June, July, September, and May ranged between 235.5 and 256.3 (*p* < 0.05), with no significant differences among them (*p* < 0.05).

Regarding monthly milk production, August showed the lowest milk yield in the early lactation period at 36.5 kg/cow/day (*p* < 0.05), followed by July, September, and October at 37.0, 37.4, and 37.0 kg/cow/day, respectively. June and May showed the highest milk yields at 38.1 and 38.9 kg/cow/day. In the mid-lactation period, August also had the lowest yield at 35.8 kg/cow/day (*p* < 0.05), while July and September followed at 37.7 and 37.4 kg/cow/day, respectively. June, May, and October had similar milk yields at 39.2, 39.0, and 39.2, respectively. In the late lactation period, milk production gradually decreased over time, with values ranging from 30.8 to 32.6 kg/cow/day (*p* < 0.05). Overall, May had the highest milk production, followed by June (*p* < 0.05).

When considering milk production adjusted for milk fat content, May had the highest yield in the early lactation period at 39.8 kg/cow/day (*p* < 0.05), while June, August, September, and October had the lowest milk yields at 28.7, 24.1, 28.1, and 28.0 kg/cow/day, respectively (*p* < 0.05). July and May had milk yields of 30.4 and 39.8, respectively. In the mid-lactation and late lactation periods, no significant differences were observed in adjusted milk production.

### 3.3. Milk Composition

The impact of high-temperature stress on the milk components of Holstein dairy cows is presented in Table 4. The milk component content did not show significant differences among the different treatments and parity stages. However, when examining milk protein specifically, it was observed that it significantly decreased from June in the early lactation period (*p* = 0.001) and the mid-lactation period (*p* = 0.022) and began to recover in October. In the late lactation period, there were no significant differences in milk protein content among the different treatments. For milk fat content, in the early lactation period, it was the highest in October. However, there were no significant differences in milk fat content during the mid-lactation or late lactation periods. The milk urea nitrogen (MUN) results showed a decrease from May to October in both the early and mid-lactation periods (*p* < 0.05). In the late lactation period, MUN decreased initially and then increased again in October (*p* < 0.05). Overall, there were no significant differences in somatic cell counts.

### 3.4. Rumen Temperature and Activity Level

The ruminal temperature and activity level are presented in Table 5. The rumen temperature was significantly higher in August at 39.1 °C during the mid-lactation period and in both August and September at 39.1 °C during the late lactation period (*p* < 0.05). However, no significant differences were observed in the early lactation period. Throughout the entire study period, there were no significant differences in activity levels between the early and mid-lactation periods, while in the late lactation period, September and October showed significantly higher activity levels at 10.0 and 10.2, respectively.

### 3.5. Correlations 

Correlations among milk production, THI values, days in milk, rumen temperature, activity, heat index, feed intake, and milk production per feed intake during the experimental period are presented in Table 6. Milk production (MP) showed significantly negative correlations with days in milk (DIM), rumen temperature (RT), and activity (*p* < 0.01) and it showed positive correlations with feed intake (FI) (*p* < 0.05) and milk production per FI (*p* < 0.01). RT showed positive correlations with minimum THI of the day (MIN), maximum THI of the day (MAX), daily percentage of under THI 72 to 78 (72 to 78), and daily percentage of under THI 78 to 89 (78 to 89) except to percentage of under THI 72 (< 72) (*p* < 0.01). The heat index (HI) tended to be positively correlated with MP, and it tended to show a negative correlation with DIM (*p* < 0.10). The THI ratio showed negative correlations with RT and activity (*p* < 0.01). Feed intake showed positive correlations with MP and the THI ratio, and it showed significantly negative correlations with MIN, MAX, 72 to 78, 78 to 89, activity, and RT (*p* < 0.01). 

The relationships among THI, MP, MP per FI, THI ratio, RT, and activity from May to October are shown in Figure 2. As the THI increased above 72, a decrease in MP was observed, and the MP recovered approximately 1 month after THI decreased below 72 (Figure 2a). As The THI changed, MP per FI was not affected by the change in the THI (Figure 2b). The RT was observed above 39.15 °C at a THI ratio under −100 (Figure 2c). In Figure 2d, a specific correlation was not found between RT and HI. 

## 4. Discussion

The climate in South Korea is gradually shifting toward a tropical climate, with a concentration of rainfall in the summer months and higher temperatures during the summer, accounting for a larger proportion of the year’s climate [14]. In particular, it has been reported that temperatures in July and August in South Korea are significantly higher than in other seasons, but recently, there have been many cases where the highest temperature in June surpasses that of July [14]. This suggests that the management of heat stress due to high temperatures is starting later. South Korea’s humidity characteristics show that July and August typically have humidity levels between 75 and 85%, while throughout the year, the humidity level ranges from 60 to 75%. The lowest humidity is reported in March and April, with levels between 50 and 70% [14]. The average humidity in this study is similar to these reports.

In the current research, the THI was calculated using the THI chart presented in NRC dairy cattle [8]. It assesses heat stress in animals using dry bulb temperature and relative humidity and defines THI values of 71 or lower as no heat stress (comfort), 72–79 as mild stress, 80–89 as moderate stress, and 90 or higher as severe stress. The temperature–humidity index is reported to be more suitable for environments with high humidity [15], and the summer climate in South Korea exhibits characteristics of a climate with very high humidity. It is considered that the THI might better reflect animal stress in South Korea because of the country’s hot and humid climate. In the current research, the average THI in August ranged from 73.6 to 81.0, indicating mild to moderate stress from June to October. The results suggest that heat stress begins in June, with the maximum THI recorded at 75.6, making June a significant month for the onset of heat stress. The data for the THI from May to October 2020 indicate that extreme levels of heat stress were not observed during the study period. Feed intake was the lowest in August, coinciding with an average THI of 77.3, which falls into the mild–moderate category, suggesting that feed intake is affected by these conditions. For the other months, the average THI remained below 72, a range known to have a lower impact on feed intake [16]. The findings regarding feed intake in the current research are consistent with other research results. However, when examining the maximum and minimum THI values for each month, it is notable that June, July, and September had THI values above 72, which are known to affect feed intake. Surprisingly, there were no significant differences in feed intake during these months. In contrast, only August had a minimum THI value exceeding 72, suggesting that the frequency and duration of THI conditions exceeding 72 are critical for affecting feed intake in dairy cows.

Milk production showed distinct patterns in the early, mid-, and late lactation periods. In the early lactation period, milk production began to decrease in July, the onset of high-temperature stress, and continued until October, with partial recovery but not to previous levels. In the mid-lactation period, October showed a complete recovery in milk production compared with earlier months. In the late lactation period, milk production decreased continuously over time, regardless of high-temperature stress, as is typical for this stage of production [8]. Overall, this study revealed that the timing of milk production recovery differed for cows in the early, mid-, and late lactation periods, with cows in the early lactation period being more vulnerable to the severe effects of high-temperature stress. This may be because cows in early lactation do not utilize the normal glucose-sparing mechanism as effectively as cows in other lactation stages during high-temperature stress conditions exceeding the threshold [17]. In summary, high-temperature stress affected milk production differently depending on the lactation stage, with the most severe impact observed in the early lactation period.

A previous study reported that an increasing THI in dairy cattle during mid-lactation negatively affects milk fat and milk protein content [18], and it was reported that when the THI exceeded 60, there was a decrease in feed intake and negative effects on milk components. Another study conducted on Holstein cows in South Korea found similar results, with a significant decrease in milk fat and milk protein as the THI increased [6]. In the current research, no significant negative impact on milk fat was observed; however, for milk protein, there was a negative impact as the THI increased, followed by recovery in October. Typically, the milk fat content is influenced by feed intake and the ratio of forage in the diet [8]. The average daily feed intake in August, 36.4 kg (22.8 kg DM), was considered sufficient for normal milk production in Holstein dairy cows and was reported to exceed the energy requirements for milk production [8].

Ruminants synthesize the most fat utilizing fatty acids produced by rumen microorganisms [8]. Extreme changes in ambient temperature can lead to alterations in the rumen microbiota [18]. On the other hand, the metabolism required to maintain the rumen temperature in ruminants decreases during high-temperature periods; however, high-temperature stress leads to a reduction in the feed intake of cows. This situation suggests that the influence of microbiota with changes in rumen temperature may be limited, while feed intake may have a greater impact. Then, ensuring sufficient energy availability in the rumen could mitigate the impact of decreased milk fat production caused by fatty acid reduction, as most of the milk fat in dairy cows is derived from rumen fatty acids [19]. On the other hand, protein synthesis is more closely linked to metabolism in the body of cattle, and it is more complex. Protein metabolism requires more energy than carbohydrate metabolism because proteins are used to build the body and need a metabolic process to excrete ammonia, a toxic substance. This is the reason why protein metabolism is more vulnerable to heat stress than carbohydrate metabolism. These phenomena may explain the results observed in milk production and protein content in the current research.

Milk urea nitrogen (MUN) is used to determine a diet’s protein ratio [20]. In lactating cows, MUN is known to increase as excess protein is supplied, and this ammonia production from excess protein is absorbed into the bloodstream and then diffuses into the mammary gland, affecting MUN levels [21]. The MUN levels in the current research, approximately 16.1 ng/mL (average for May to June), were within the normal range for Holstein cows in South Korea [22,23]. MUN for optimal production is recommended at <20 mg/dL, with a threshold value of 14 mg/dL for optimal reproduction [24]. As feed intake reduces because of heat stress, it causes a reduction in the energy and protein supplies in the rumen. This situation can decrease nitrogen metabolism by the rumen microbes and reduce ammonia production [25]. This situation may lead to decreased MUN content, as observed in the current research. 

Milk production was shown to decrease with stress once the THI value was above 72 (Figure 2a). During the experimental period of this study, when a decrease in milk production was observed, the THI ranged from 72 to 81. During this time, milk production decreased by approximately 2%. The decrease in milk production was shown to recover approximately one month after the THI fell below 72. The effects of heat stress on milk yield are very diverse and generally result from decreased feed intake to relieve body heat accumulation [26]. Generally, dairy cattle tend to maintain a core body temperature higher than the ambient temperature [27]. With increasing THI levels, when the ambient temperature exceeds the core body temperature, cows increase respiration and heat excretion. When the THI threshold is exceeded, these mechanisms fail to prevent excess heat accumulation, limiting cows’ ability to maintain homeothermy and exposing them to heat stress [27]. Recovery from elevated heat shock protein levels due to high-temperature stress is reported to occur 6 to 10 days after returning to normal temperatures [28]. Although the recovery of protein levels occurred within 10 days in a previous study [28], milk production in the current research recovered after a month. Furthermore, feed intake rapidly recovered to levels observed before reaching THI threshold conditions. This indicates that recovering productivity takes more time than recovering at the cellular level. In terms of milk production per feed intake, there was no immediate decrease in milk production during the high-temperature stress period. However, after passing the stress period, as the THI recovered, a subsequent decrease in milk production was observed (Figure 2b). This is an example that indirectly indicates damage at the cellular level in dairy cows. Rumen temperature is regarded as a reliable indicator of core body temperature, as is rectal temperature [29]. In the current research, the THI ratio was negatively correlated with the rumen temperature estimated by the sensor (−0.774; Table 6). In the moderate THI condition, the rumen temperature was observed over 39.15, and the THI ratio value was under minus 100. In the current research, a specific case was observed in which, when the THI ratio was below −100, the rumen temperature did not exceed 39.15 °C. In such cases, the THI was below 72 and was maintained for over 30% of the time on the day before and the day after. Additionally, it was shown that THI values above the threshold affected the rumen temperature only after persisting for more than 2 days. In the current research, the rumen temperature was closely associated with a period of reduced milk yield. This study did not collect data from flow meters or surveillance cameras, so individual water intake was not calculated. However, in the current research, the rumen temperature data involved changes associated with water intake data. A rumen insertion sensor provides information on water intake frequency based on variations in rumen temperature. It is well known that heat stress increases water consumption, and depending on the temperature of the water provided to the animals, increased water intake could significantly affect ruminal temperature [30,31]. In the current research, the rumen sensor data indicated that the rumen temperature dropped by approximately 1 degree when water was consumed and then recovered within a few seconds. This suggests that the impact of water intake on the rumen temperature measurements was not detected when using the rumen sensor. This could explain why negative changes in rumen temperature due to water intake were not observed in the current research, unlike in other studies. Therefore, we consider that a rumen temperature of 39.15 °C could serve as an indicator of stress in dairy cows. Based on our results, we suggest that this value could be used as a foundation for developing technologies to manage high-temperature environments for dairy cows.

## 5. Conclusions

The effects of heat stress on the productivity of dairy cows were evaluated in the current research. During the experimental period, the THI values ranged from 46.9 to 81.0, with the most heat stress occurring in August. In the current research, a decrease in dairy cow productivity was observed over the THI threshold of 72. It took about one month for milk production to recover after the high-temperature period ended. The THI ratio calculated based on the daily THI distribution showed a strong negative correlation with rumen temperature (−0.744; *p* < 0.001). During the period when a significant decrease in productivity due to high-temperature stress was observed, rumen temperatures were higher than 39.15 °C. Based on our results, we suggest that rumen temperature and THI distribution of the day could be used as foundational data for developing guidelines to manage high-temperature environments for dairy cows.

## Figures and Tables

**Figure 1 animals-14-02848-f001:**
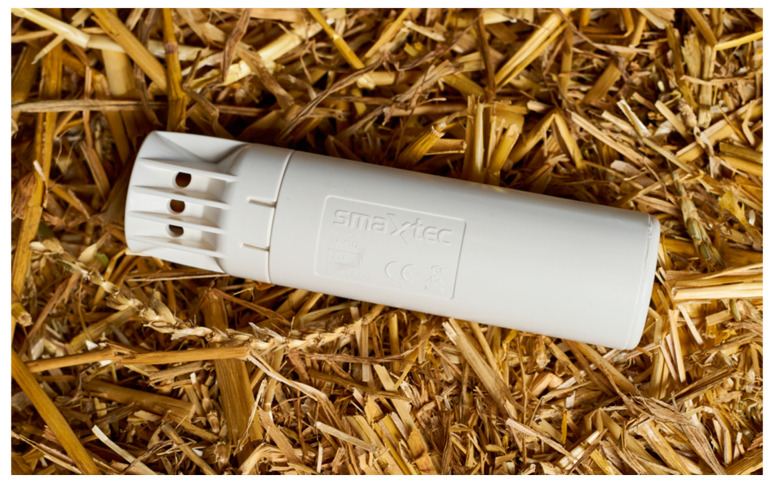
Rumen insert-type sensor (Smaxtec pH plus bolus).

**Figure 2 animals-14-02848-f002:**
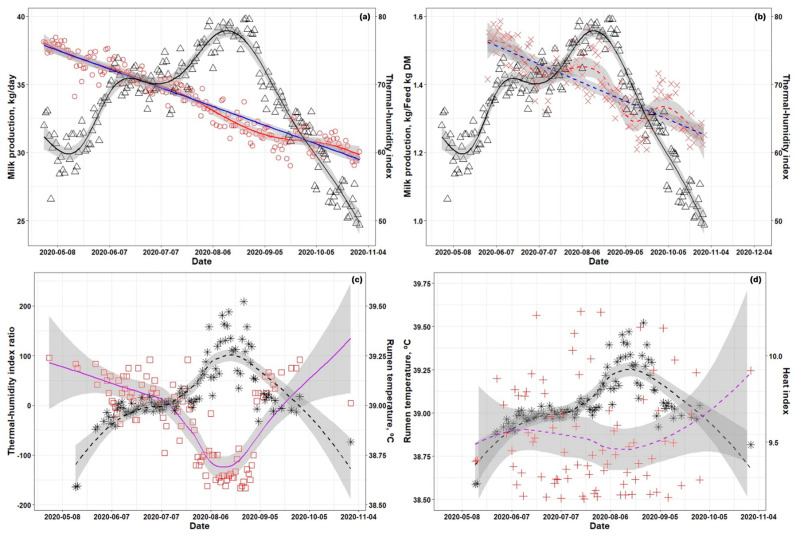
Relationships among the temperature–humidity index (THI), milk production (MP), milk production per feed intake (MPFI), THI ratio, rumen temperature (RT), and heat index (HIduring the experimental periods. (**a**), THI and MP changes from May to October; (**b**), THI and MPFI changes from May to October; (**c**), THI ratio and RT changes from May to October; (**d**), RT and HI changes from May to October. ○, MP; △, THI; ×, MPFI; *, RT; □, THI ratio; +, heat index. Red line, the MP trend line was analyzed using generalized additive models; blue line and blue dotted line, the MP trend line was analyzed using linear regression; red dotted line, the MP per feed intake trend line was analyzed using locally estimated scatterplot smoothing (loess); black line and black dotted line, the THI and RT trend line was analyzed using the loess method. Purple line and purple dotted line, the THR ratio and the heat index trend line were analyzed using the loess method. The THI ratio was calculated as follows: THI ratio = 1 × Daily percentage under THI 72 − 1 × Daily percentage under THI 72 to 78 − 2 × Daily percentage under THI 78 to 89. The slope weights for the THI ratio calculation were set arbitrarily regarding the correlation with milk production.

**Table 1 animals-14-02848-t001:** Chemical composition of the experimental diet.

Items	TMR
DM, %	62.67
CP, %DM	18.21
EE, %DM	3.51
Ash, %DM	8.50
NDF, %DM	46.61
eNDF, %DM	33.00
ADF, %DM	26.39
NFC, %DM	23.18
TDN, %DM	72.99

TMR, total mixed ration; DM, dry matter; CP, crude protein; EE, ether extract; NDF, neutral detergent fiber; eNDF, effective neutral detergent fiber ADF, acid detergent fiber; NFC, non-fiber carbohydrate; TDN, total digestible nutrients.

**Table 2 animals-14-02848-t002:** Temperature, relative humidity, and the THI of the experimental farm by lactation stage from May to October (n = 125).

Items	May	June	July	Aug.	Sep.	Oct.	SEM	*p*-Value
	Temperature, °C							
Average	16.8 ^e^	22.1 ^c^	22.9 ^b^	26.2 ^a^	20.3 ^d^	13.3 ^f^	0.04	<0.001
Minimum	12.3 ^e^	18.3 ^c^	20.0 ^b^	23.8 ^a^	16.5 ^d^	7.8 ^f^	0.21	<0.001
Maximum	22.3 ^d^	27.4 ^b^	26.9 ^b^	29.5 ^a^	25.1 ^c^	19.6 ^e^	0.22	<0.001
	Relative humidity, %							
Average	73.6 ^d^	78.3 ^c^	82.2 ^b^	86.3 ^a^	78.4 ^c^	68.7 ^e^	0.72	<0.001
Minimum	48.2 ^b^	52.7 ^b^	63.7 ^a^	71.8 ^a^	54.6 ^b^	37.7 ^c^	1.60	<0.001
Maximum	93.3 ^abc^	94.8 ^ab^	95.8 ^a^	94.8 ^ab^	92.9 ^bc^	90.9 ^c^	0.54	0.002
	Temperature humidity index							
Average	61.1 ^e^	69.6 ^c^	71.6 ^b^	77.3 ^a^	67.1 ^d^	55.8 ^f^	0.18	<0.001
Minimum	54.3 ^cd^	64.8 ^ab^	67.6 ^ab^	73.6 ^a^	61.6 ^bc^	46.9 ^d^	1.71	<0.001
Maximum	68.2 ^d^	75.6 ^b^	76.1 ^b^	81.0 ^a^	73.0 ^c^	64.7 ^e^	0.19	<0.001

THI, temperature–humidity index; Aug., August; Sep., September; Oct., October; SEM, standard error of the mean. ^a–f^ Means in the same row with different superscripts differ significantly at *p* < 0.05.

**Table 3 animals-14-02848-t003:** Dry matter intake and milk production of lactating dairy cows by lactation stage from May to October (n = 125).

Items ^1^	May	June	July	Aug.	Sep.	Oct.	SEM	*p*-Value
Intake								
DMI, kg/head/d	NM	23.8 ^a^	24.0 ^a^	22.8 ^b^	23.8 ^a^	23.5 ^a^	0.15	<0.001
	Days in milk, d							
Early lactation	37.5 ^c^	43.6 ^a^	38.4 ^c^	31.7 ^d^	39.9 ^bc^	42.0 ^ab^	0.70	<0.001
Mid-lactation	104.7 ^b^	106.3 ^b^	106.4 ^b^	109.7 ^a^	105.5 ^b^	95.1 ^c^	0.76	<0.001
Late lactation	235.3 ^c^	238.6 ^c^	253.6 ^b^	259.2 ^b^	256.3 ^b^	275.6 ^a^	1.49	<0.001
	Milk production, kg/cow/day							
Early lactation	38.9 ^a^	38.1 ^ab^	37.0 ^bc^	36.5 ^c^	37.4 ^bc^	37.0 ^bc^	0.39	<0.001
Mid-lactation	39.0 ^a^	39.2 ^a^	37.7 ^b^	35.8 ^c^	37.4 ^b^	39.2 ^a^	0.26	<0.001
Late lactation	32.6 ^a^	32.2 ^ab^	31.9 ^b^	31.0 ^c^	31.4 ^c^	30.8 ^c^	0.13	<0.001
	4%FCM ^2^, kg/cow/day							
Early lactation	39.8 ^a^	28.7 ^b^	30.4 ^ab^	24.1 ^b^	28.1 ^b^	28.0 ^b^	3.63	<0.001
Mid-lactation	49.4	50.6	49.2	45.9	50.2	48.5	5.93	0.727
Late lactation	40.4	39.2	38.7	36.8	36.2	34.4	1.57	0.093
	FPCM ^3^, kg/cow/day							
Early lactation	47.6	46.8	51.1	47.6	48.2	43.6	7.93	0.399
Mid-lactation	47.7	48.6	46.7	43.7	47.3	46.2	5.36	0.553
Late lactation	38.9	38.3	37.2	35.6	35.2	33.9	1.55	0.124

Aug., August; Sep., September; Oct., October; SEM, standard error of the mean; DMI, dry matter intake; NM, not measured; 4%FCM, 4% fat corrected milk; FPCM, fat-protein corrected milk. ^1^ Early lactation, 10 to 70 days (n = 75); mid-lactation, 71 to 140 days (n = 80); late lactation, over 140 days (n = 286); ^2^ 4%FCM = 0.4 × milk production + 15 × milk fat yield; ^3^ FPCM = milk production × (0.337 + 0.116 × milk fat (%) + 0.06 × milk protein (%). ^a–c^ Means in the same row with different superscripts differ significantly at *p* < 0.05.

**Table 4 animals-14-02848-t004:** Milk compositions of lactating dairy cows by lactation stage from May to October (n = 75).

Items ^1^	May	June	July	Aug.	Sep.	Oct.	SEM	*p*-Value
	Fat, %							
Early lactation	5.7	5.5	5.8	6.6	5.9	5.5	0.74	0.311
Mid-lactation	4.9	4.9	5.4	4.9	5.5	5.2	0.29	0.591
Late lactation	5.3	5.1	5.2	5.1	5.1	5.0	0.22	0.942
	Protein, %							
Early lactation	3.2 ^a^	3.1 ^ab^	3.1 ^ab^	3.0 ^ab^	2.8 ^b^	3.2 ^a^	0.11	0.001
Mid-lactation	3.3 ^a^	3.2 ^ab^	3.2 ^ab^	3.1 ^b^	3.0 ^b^	3.2 ^ab^	0.07	0.022
Late lactation	3.2	3.4	3.2	3.2	3.3	3.5	0.12	0.348
	Solid not fat, %							
Early lactation	8.78 ^ab^	8.78 ^ab^	8.70 ^ab^	8.68 ^ab^	8.56 ^b^	8.99 ^a^	0.16	0.036
Mid-lactation	9.00	8.77	8.69	8.76	8.72	8.88	0.08	0.068
Late lactation	8.54	9.00	8.45	8.56	8.86	8.87	0.26	0.589
	Milk urea nitrogen, ng/mL							
Early lactation	16.3 ^a^	12.8 ^bc^	14.4 ^ab^	12.04 ^bc^	10.8 ^c^	10.8 ^c^	0.62	0.001
Mid-lactation	17.2 ^ab^	16.1 ^abc^	15.5 ^bcd^	14.3 ^d^	12.2 ^cd^	13.4 ^cd^	0.58	0.001
Late lactation	17.3 ^a^	17.1 a	15.2 ^ab^	14.0 ^bc^	12.6 ^c^	13.0 ^bc^	1.27	0.001
	Somatic cell counts, 10^3^ cell/mL							
Early lactation	203.4	179.4	187.4	330.5	169.2	121.2	93.65	0.263
Mid-lactation	103.3	125.6	141.1	132.0	314.2	104.9	73.32	0.434
Late lactation	145.7	191.2	227.9	271.5	261.0	203.8	47.77	0.475

SEM, standard error of the mean; Aug., August; Sep., September; Oct., October. ^1^ Early lactation, 10 to 70 days (n = 75); mid-lactation, 71 to 140 days (n = 80); late lactation, over 140 days (n = 286). ^a–d^ Means in the same row with different superscripts differ significantly at *p* < 0.05.

**Table 5 animals-14-02848-t005:** Rumen temperature and activity of lactating dairy cows by lactation stages from May to October (n = 125).

Items ^1^	May	June	July	Aug.	Sep.	Oct.	SEM	*p*-Value
	Rumen temperature, °C							
Early lactation	36.6	39.0	39.1	39.3	39.3	38.8	0.70	0.118
Mid-lactation	38.7 ^d^	38.9 ^bcd^	38.9 ^bc^	39.1 ^a^	38.9 ^ab^	38.7 ^cd^	0.06	0.001
Late lactation	38.7 ^d^	38.8 c^d^	38.9 ^b^	39.1 ^a^	39.1 ^a^	38.8 ^bc^	0.04	0.001
	Activity							
Early lactation	10.0	9.0	9.3	10.4	10.1	9.8	0.70	0.731
Mid-lactation	9.1	9.7	9.1	9.4	9.7	10.8	0.58	0.179
Late lactation	9.1 ^b^	8.8 ^b^	9.3 ^b^	9.6 ^ab^	10.0 ^a^	10.2 ^a^	0.33	0.036

SEM, standard error of the mean; Aug., August; Sep., September; Oct., October. ^1^ Early lactation, 10 to 70 days (n = 75); mid-lactation, 71 to 140 days (n = 80); late lactation, over 140 days (n = 286). ^a–d^ Means in the same row with different superscripts differ significantly at *p* < 0.05.

**Table 6 animals-14-02848-t006:** Correlations between milk production, THI values, days in milk, rumen temperature, activity, heat index, feed intake, and milk production per feed intake during the experimental period (n = 125).

Items	Min	Max	<72	72 to 78	78 to 89	DIM	RT	Activity	HI	THI Ratio ^1^	FI
MP	0.106	0.117	0.089	−0.023	−0.123	−0.986 ***	−0.417 ***	−0.786 ***	0.146 *	0.092	0.226 **
MIN		0.875 ***	−0.792 ***	0.757 ***	0.570 ***	−0.152	0.724 ***	0.128	0.005	−0.797 ***	−0.213 **
MAX			−0.774 ***	0.740 ***	0.632 ***	−0.240	0.665 ***	0.129	0.082	−0.807 ***	−0.232 **
<72				−0.896 ***	−0.718 ***	−0.333	−0.730 ***	−0.310 ***	−0.028	0.986 ***	0.316 ***
72 to 78					0.385 ***	0.333	0.646 ***	0.262 ***	0.027	−0.849 ***	−0.216 **
78 to 89						NA	0.594 ***	0.228 **	−0.017	−0.809 ***	−0.355 ***
DIM							0.312	0.757 *	−0.757 *	−0.333	−0.711
RT								0.586 ***	−0.077	−0.744 ***	−0.385 ***
Activity									0.069	−0.303 ***	−0.205 *
HI										−0.014	0.063
THI ratio											0.344 ***

MP, milk production; Min, minimum THI of the day; Max, maximum THI of the day; <72, daily percentage under THI 72; 72 to 78, daily percentage under THI 72 to 78; daily percentage under THI 78 to 89; DIM, days in milk; RT, rumen temperature; HI, heat index; FI, feed intake. ^1^ The THI ratio was calculated as THI ratio = 1 × Daily percentage under THI 72 − 1 × Daily percentage under THI 72 to 78 − 2 × Daily percentage under THI 78 to 89. The slope weights for the THI ratio calculation were set arbitrarily regarding the correlation with milk production. Superscripts mean * *p* < 0.10, ** *p* < 0.05, and *** *p* < 0.01.

## Data Availability

No new data were created or analyzed in this study. Data sharing is not applicable to this article.

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
