# Peer review of "Effect of the Temperature–Humidity Index on the Productivity of Dairy Cows and the Correlation between the Temperature–Humidity Index and Rumen Temperature Using a Rumen Sensor"

_animals, 2024, doi:10.3390/ani14192848_

Round 1

Reviewer 1 Report

Comments and Suggestions for Authors

The subject of heat stress on dairy cattle is always an important one. Even though the results may apply particularly to a country or region within this country, the information supplied could be on interest.

Regretfully, the description of the materials and methods lack specificity, and it makes it very difficult to assess the soundness of the analyses performed in this study,

In general, it is not clear what kind of information was used, was it daily, monthly? Nor the variables measured and used. For example, days dry is not mentioned until the results section. Stating the number of observations per period and per variable would help to understand this. Data loggers have a measure every 30 mins. How were these used?

The methods section must be improved to understand the actual analyses performed. The analysis of milk yield cannot be fully understood until it is described how DIM was handled. It seems that the study evaluates cows in different days in milk so, when the milk yields are evaluated by month (Table 3) cows in June for example would have one more month in milk, and it would be in a different stage of lactation. How was this handled? Also, cows that initiated the study as “early lactation” should have ended it at mid-lactation at minimum. Was this allowed?  

The impact of heat stress on dairy cattle is widely known and not only the severity but the duration of the heat stress (days over 72 ITH) has to be looked at. This paper will benefit from looking at that information to obtain better results.

Line 49.               Retention rate. Can you explain what is meant by retention rate?

Line 67-69.         Please mention what is measured by each device. This would help understand lines 69-71.

Line 75-76.         “the rumen sensor” is only one kind of rumen sensor? Being this the objective of the paper it demands more precision.

Lines 90-92.       What is the effect of the fans. Is there information on the conditions before using the fans. More information on the fan use is needed. At what hours were the fans turned on? Turned off? What was the temperature during these hours?

Lines 99-100.     Can you explain what is meant by “at the height of the cows”? Does this mean each cow had a data logger?

Line 107.            “THI represents the THI” is not needed.

Lines 109-116.   These levels and its description is presented elsewhere, thus this part is better suited for the introduction.

Lines 134-135.   The “motion detecting sensors” are not defined. Are they within the “Bolus oH plus”? Are they additional sensors? Please describe.

Lines 151-155.   The model needs to be revised. The model stated does not include the random effect. When a random effect is included, the assumed distribution and the variance covariance matrix needs to be defined. The experimental periods are not defined, are these the calendar months? The model as presented assumes that THI is a discrete variable. Is this true? And if it is, are the levels the ones described in lines 109-116? How were days in milk included in the model? Was milk adjusted for EM?

Line 165.            Change “was” for “were”

Line 166.            The term “THI ratio” is included but has not been defined.

Lines 160-167.   In general, the explanation of this part of the methodology is difficult to understand Need rewording.

Line 198.            Change “Daays” in the title

Lines 208-216.   The concept of early, mid and late lactation has not been mentioned nor defined.

Line 251.            Review the English please

Lines 277-286.   The problem persists with the THI ratio.

Line 288.            Does it mean Fig 2?

Line 322-323.    It is not clear why “THI may be more suitable for evaluating heat stress….”

Line 339-352.    The whole milk yield discussion is pendant on the definition of how the data was handled.

Figure 2.             What was the analysis applied to the MP and other variable trends? There are no regression coefficients reported in the paper.

Line 391.            There is a reference to Figure 2a. but Figure 2 does not show identifiers for the four graphs.

Line 391.            What was implied by a “decreased flow rate”? Were flow rates measured in these cows?

Line 434.            Perhaps the results could be used to develop  methods to manage heat stress more than “technologies”?

Comments on the Quality of English Language

The paper needs English revisions

Author Response

Thank you for your kind review.
I have written a response to your review in the uploaded file.

Reviewer 2 Report

Comments and Suggestions for Authors

First, the topic of the manuscript is very relevant in the current context of dairy industry and the respective study has very strong merits, namely the sampling dimensions (e.g. 75 milking cows with sensors), the extensive list of parameters analyzed and the temporal duration. These features guaranteed comprehensive and reliable results at a statistical level.

Although there are not many novelties in the study's conclusions, it undoubtedly reinforces the importance of environmental control to ensure animal welfare and adequate production levels.

However, the manuscript still has room for improvement if some aspects are considered. The most relevant is the water consumption of animals with sensors in the rumen, which will be mentioned later in this review.

To begin with, the title could be improved as it is a bit vague, not very specific.

Lines 41-43, I recommend more rigor regarding references to the Holstein breed. Today, the term "Holstein" is used to describe the North American breed and the use of that breed in Europe (primarily through the importation of semen). Holstein cattle are descended from the Friesians that came with settlers to North America and were selected there for higher production levels (they also increased their body dimensions) due to the greater availability of resources on American farms and probably the bigger demand for dairy products. "Friesian" is used to describe animals of traditional European descent, of smaller body dimensions, that are still raised for both dairy use and meat.

Lines 50- 51, the statement seems very definitive, and I doubt that the percentage referred to is always this, especially due to the involvement of many other factors.

Lines 58-60, elucidate the mechanisms of the following sentence, as the relationships are not very clear: “To maintain normal CO2 levels, bicarbonate (HCO3-) secretion in the kidneys decreases, leading to decreased dry matter intake and reduced chewing activity.”

Lines 73-75, I suggest mentioning here what the ruminal sensor measures: rumen temperature and activity. There are already sensors that measure rumen pH, for example.

Genetic improvement of dairy cattle has focused on increasing milk production, which has turn cows into metabolic "heat producing machines", making them highly susceptible to warm climates. Thus, the authors could have adjusted for production levels (e.g. with determination of the percentual impact of heat stress on production levels) but instead opted for categorization into early, mid or late lactation, which also may be acceptable.

In the Discussion, and in relation to figure 2, it would enrich the manuscript if water intake was addressed in the two graphs relating to rumen temperature, as it highly influences the latter.

It is well known that heat stress increases water consumption (e.g. McDonald, et al. Hot weather increases competition between dairy cows at the drinker. J Dairy Sci 2020; 103(4):3447-3458.) and, depending on the temperature of the water provided to animals, ruminal temperature could be significantly affected by increased water consumption.

From what was read, this parameter was not monitored in this study, which is understandable as there were 75 cows and its determination would involve individual stalls, flow meters, surveillance cameras, etc. In any case, there is significant information about water consumption and heat stress in the literature that could be considered to help interpret the results.

In this sense, some articles I found in a quick search, but there are more:

Wilks et al. Responses of lactating Holstein cows to chilled drinking water in high ambient temperatures. J Dairy Sci 1990; 73:1091-1099.

Brod et al. Effect of water temperature on rumen temperature, digestion and rumen fermentation in sheep. J Anim Sci 1982; 54(1):179-82.

Grossi S, et al. Effects of Heated Drinking Water on the Growth Performance and Rumen Functionality of Fattening Charolaise Beef Cattle in Winter. Animals (Basel) 2021; 11(8):2218.

Author Response

(The authors gave the same response as above.)

Reviewer 3 Report

Comments and Suggestions for Authors

Dear Authors,

Your manuscript titled 'Evaluation of heat stress on the milk productivity using rumen sensor in dairy cows' is addressing an interesting topic of research. It might be suitable for publication in Animals under major revision. Please, see below a list of comments/suggestions to be applied by you for improving the quality of the manuscript before accepting it to be published.

Yours sincerely,

Reviewer.

L8 Link the author/s with the institution.

L14-L15 Replace 'cor-relation' by 'co-rrelation'.

L30 Replace 'remen' by 'rumen'.

L45-46 Replace 'rap-idly' by 'ra-pidly'.

L81 Add a general description of the animals under study. Provide information related to breed, average parity, days in milk, body weight and body condition score, etc. at the beginning of the study.

L84 Describe how dry matter intake was measured.

L84-85 Write in italics 'ad libi-tum'.

L90 Add the dates when fresh sawdust was incorporated.

L99-L100 Replace 'at-tached' by 'a-ttached'.

L127-L128 Replace 'As-sociation' by 'A-ssociation'.

L132 Explain the criteria used for selection of the 75 cows in which the rumen sensor was inserted into.

L149-L159 Rewrite the Statistical Analysis section. Recheck the model. Explain which were the experimental treatment groups created and how did you create them. Give details about the lactation period. How did you create the three 3 categories (early, mid and late)? Explain why this factor isn't represented in your statistical model and Tables 3-5 are considering lactation period as a factor of your study.

L172 Replace 'was highest' by 'was the highest'.

L176 Replace 'highest' by 'maximum'.

L179 Replace 'was highest' by 'was the highest'.

L186 Replace 'was highest' by 'was the highest'.

L188 Replace 'was in' by 'was found in'.

L189 Replace 'and May' by 'while May'.

L190 Replace 'was in' by 'was found in'.

L191 Replace 'and September' by 'while September''.

L198 Replace 'Daays' by 'Days'.

L202 Replace 'and August' by 'while August'.

L208 Replace 'yield' by 'milk yield'.

L211 Replace 'had' by 'showed'.

L213 Replace 'yields' by 'milk yields'.

L215-K216 Replace 'produc-tivity' by 'produ-ctivity'.

L225-L226 Replace 'mat-ter' by 'ma-tter'.

L239 Replace 'was highest' by 'was the highest'.

L267-L268 Replace 'cor-relation' by 'co-rrelation'.

L280 Replace 'minimum' by 'maximum'.

L283 Replace 'calcualted' by 'calculated by'.

L289 Replace 'Fig-ure' by 'Fi-gure'.

L292 Replace 'did not show' by 'was not found'.

L316 Remové 'was'.

L317 Replace 'In this study' by ' In the current research'.

L329 Replace 'was lowest' by 'was the lowest'.

L345-L346 Replace 'dif-fered' by 'di-ffered'.

L377-L378 Replace 'produc-tivity' by 'produ-ctivity'.

L393-L394 Replace 'ap-proximately' by 'a-pproximately'.

L404-L405 Replace 'recov-ered' by 'reco-vered'.

L422-L423 Replace 'temper-ature' by 'tempe-rature'.

L429 Delete a space.

L433-L434 Replace 'sug-gested' by 'su-ggested'.

L450-L504 Review all referentes and cite them according to Animals' instructions for authors.

Author Response

(The authors gave the same response as above.)

Round 2

Reviewer 3 Report

Comments and Suggestions for Authors

Dear Authors,

I'm glad to inform you that your manuscript entitled "Effect of temperature-humidity index on the productivity of 2 dairy cows and the correlation between temperature-humidity 3 index and rumen temperature using rumen sensor" is now acceptable for publication in present form at Animals.  Congratulations to all of you for this achievement!

Yours sincerely,

Reviewer.

Author Response

Thank you for your kind review.